# New Approaches to Increasing the Superhydrophobicity of Coatings Based on ZnO and TiO₂

**Arsen E. Muslimov** [1,*] , **Makhach Kh. Gadzhiev** [2] **and Vladimir M. Kanevsky** [1]

1    Shubnikov Institute of Crystallography, Federal Scientific Research Centre "Crystallography and Photonics", Russian Academy of Sciences, 119333 Moscow, Russia; kanevsky@crys.ras.ru
2    Joint Institute for High Temperatures, Russian Academy of Sciences, 125412 Moscow, Russia; makhach@mail.ru
*    Correspondence: amuslimov@mail.ru

**Abstract:** The work presented is devoted to new approaches to increasing the superhydrophobic properties of coatings based on zinc oxide (ZnO) and titanium dioxide (TiO₂). There is an innovation in the use of inorganic coatings with a non-polar structure, high melting point, and good adhesion to ZnO, in contrast to the traditionally used polymer coatings with low performance characteristics. The maximum superhydrophobicity of the ZnO surface (contact angle of 173°) is achieved after coating with a layer of hematite (Fe₂O₃). The reason for the abnormally high hydrophobicity is a combination of factors: minimization of the area of contact with water (Cassie state) and the specific microstructure of a coating with a layer of non-polar Fe₂O₃. It was shown that the coating of ZnO structures with bimodal roughness with a gold (Au) layer that is 60-nm thick leads to an increase in the wetting contact angle from 145° to 168°. For clean surfaces of Au and hematite Fe₂O₃ films, the contact angle wets at no more than 70°. In the case of titanium oxide coatings, what is new lies in the method of controlled synthesis of a coating with a given crystal structure and a level of doping with nitrogen using plasma technologies. It has been shown that the use of nitrogen plasma in an open atmosphere with different compositions (molecular, atomic) makes it possible to obtain both a hydrophilic (contact angle of 73°) and a highly hydrophobic surface (contact angle of 150°).

**Keywords:** zinc oxide; titanium dioxide; rutile; nitrogen; Au; hematite; superhydrophobic; hydrophilic; photocatalytic reaction

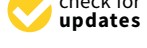

## 1. Introduction

Oxides of zinc (ZnO) and titanium, due to their unique physicochemical properties, as well as low cost and low toxicity, are today the most studied oxide materials. Devices based on zinc and titanium oxides are used in various industries: from modern electronics and photonics devices, operated under a range of conditions (low and high temperatures and pressures, in various gases, etc.) to bioanalytical devices and biosensors. Special attention is paid to the possibility of using zinc and titanium oxides in hydrogen power engineering. In this regard, one of the principal tasks is the achievement of high moisture resistance and resistance to biofouling and pollution of an inorganic and organic nature. An effective way to achieve this stability is to increase the hydrophobic properties of the surface. In the normal state, a zinc oxide (wurtzite crystal structure) surface is highly hydrophilic (wetting contact angle less than 5°) and even prolonged annealing at 350 °C does not lead to noticeable changes. [1] This is due to the characteristics of the interaction of water molecules with a ZnO surface [2]: a strong covalent bond between oxygen of the water molecule and the surface of the zinc atom, as well as the formation of O-H groups between the hydrogen of the water molecule and the surface oxygen atoms. As for the properties of a TiO₂ surface, this is more complicated. TiO₂ is characterized by polymorphism and can exist in three modifications: anatase, brookite, rutile [3]. Rutile is the stable high-temperature phase. The general trend, according to [4], is as follows: water

molecules on the surface of titanium dioxide dissociate with the formation of OH and H groups, which form bonds with the surface atoms of titanium and oxygen, respectively. The results [5] demonstrate the following feature of the interaction of $TiO_2$ with water molecules: a monophase surface characterized as highly hydrophobic, and in the case of a mixed (anatase, rutile) composition, a hydrophilic surface. This was not understood for a long time, until it was shown in [6] that the high hydrophobicity of the crystalline monophase of titanium dioxide can be associated with the formation of an ordered molecular monolayer of carboxylic acids on the surface.

According to the classical Young equation [7], to enhance the hydrophobicity of a solid surface, it is necessary to reduce its free surface energy at the three-phase contact point. We use generally accepted definitions of states which consider the contact angle $\theta$: hydrophilic ($\theta < 90°$), hydrophobic ($\theta > 90°$), and superhydrophobic ($\theta > 150°$). Due to the particularities of the crystal structure, ZnO is characterized by polarity and, as a consequence, increased surface energy. Traditionally, the hydrophobic state of a ZnO surface is achieved by the formation of specific structures such as ensembles of microrods [8] to minimize the area of contact with the liquid. To achieve record values of superhydrophobicity of ZnO (contact angle close to 170°), it is necessary to reduce the dipole–dipole interaction with the water film; for example, by covering with a layer of non-polar polymer material [9]. However, polymer coatings have a relatively low melting point, which significantly reduces the performance of devices based on superhydrophobic ZnO. In particular, the Teflon-AF grade (DuPont, Wilmington, DE, USA) used in [9] has a decomposition temperature of 360 °C. Complex and expensive equipment is required to obtain a polymer coating with good adhesion to the sample surface and thermal stability. On the other hand, complexes of materials possessing high functionality, in particular, $ZnO/Au$, $ZnO/Fe_2O_3$, are effective selective gas sensors and photocatalysts [10]. In this work, it is proposed to coat ZnO with non-polar inorganic materials with high heat resistance and good adhesion to ZnO, for example Au and $Fe_2O_3$, to increase superhydrophobicity. Superhydrophobic catalysts can have enhanced catalytic activity when introducing an air layer between the catalyst and liquid. The electrons from the reacting interface are transferred fast to radicals by acceptors such as oxygen to minimize the electron–hole recombination. Furthermore, gold contacts are traditionally used in microelectronics, and increasing the hydrophobicity in the Au-ZnO system can increase the life of devices.

Disputes about the hydrophilic properties of Au have been going on for a long time [11], and most researchers tend to consider the clean surface of gold to be weakly hydrophilic (wetting contact angle 30–80°). Perhaps this is due to the weak Au-H hydrogen bond [12], which plays an important role in the interaction of water molecules with the gold surface. It is well known that minor carbon contamination, even less than a monolayer, increases the hydrophobicity of a gold surface. As for hematite $Fe_2O_3$, according to the experimental [13] and calculated data [14], the contact wetting angles are 53° and 60°, respectively. Thus, standard Au and $Fe_2O_3$ coatings are weakly hydrophilic compared to ZnO.

$TiO_2$ coatings with desired properties can be synthesized by directly controlling the synthesis process. Titanium is less volatile than zinc, which makes it possible to first form a coating of pure titanium and then process it in plasma in an open atmosphere [15]. Since the composition of the plasma depends on temperature [16], it is possible to influence the structural-phase composition of the oxide coating formed. For greater control over the oxidation and solid phase crystallization process, nitrogen plasma treatment in an open atmosphere can be used. Additionally, doping with nitrogen affects the hydrophobic properties of a $TiO_2$ coating [17].

The work presented here is devoted to new approaches to improving the superhydrophobic properties of coatings based on ZnO and $TiO_2$ (Figure 1). The effect of inorganic coatings (Au, $Fe_2O_3$) with a non-polar structure, high melting point and good adhesion to ZnO on the superhydrophobic properties of ZnO coatings is investigated. The possibility

of controlled synthesis of a TiO$_2$ coating with a given crystal structure, nitrogen doping level and hydrophobic properties is considered.

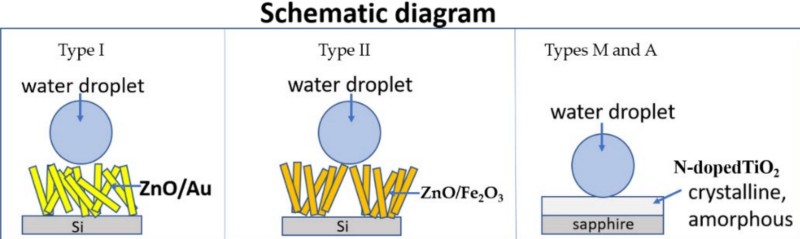

**Figure 1.** Schematic diagram.

## 2. Materials and Methods

### 2.1. Preparation of ZnO/Au, ZnO/Fe$_2$O$_3$ Samples

ZnO samples with bimodal roughness (sample type I) of the surface on sapphire substrates were formed by catalytic synthesis from the gaseous phase. The deposition of ZnO was carried out in the growth zone from the gaseous phase, consisting of zinc vapor and a gas mixture of argon and oxygen (oxygen concentration 10%). The reactor (Lab-term, Moskov, Russia) was preliminarily evacuated to a pressure of 10 Pa. Silicon wafers were used as the substrates. Synthesis conditions: temperature of the evaporation zone (T$_1$ = 630 °C); temperature of the growth zone (T$_2$ = 580 °C) and synthesis time 20 min. A feature of the synthesis was a sharp decrease in temperature in the zone of zinc evaporation 17 min after the start of the synthesis. Then, on a part of the surface of the ZnO sample (type I), an Au layer with a thickness of about 60 nm was deposited in a VN-2000 thermal spraying device (AcademPribor, Moskov, Russia). The vacuum efficiency was 10$^{-6}$ Pa. Additionally, a test sample of a continuous Au film (100 nm thick) on sapphire was grown.

Conditions for the synthesis of type II ZnO samples: an ordered Au structure was preliminarily formed in the form of square cells with a period of 8 μm, on which, under the above conditions (in the mode of the type I sample), a ZnO microstructure was formed. The deposition of iron (thickness of the order of 100 nm) on a part of a type II sample was carried out by magnetron sputtering. To form a Fe$_2$O$_3$ coating, the type II sample was annealed in atmospheric conditions in an oven at a temperature of 700 °C for 2 h. The thickness of the deposited metal layers was controlled using a KIT-1 quartz thickness gauge (AcademPribor, Moskov, Russia).

Silicon (111) was used as substrates for the formation of the ZnO microstructure on which all technological processes were undertaken. Since the formation of the ZnO microstructure was not influenced by the substrate, the substrate material was not of fundamental importance.

### 2.2. Preparation of TiO$_2$(N) Samples

To obtain TiO$_2$ samples, a titanium film (500 nm thick) was preliminarily deposited on a sapphire substrate by magnetron sputtering, after which the samples were treated with a flow of low-temperature high-enthalpy nitrogen plasma in an open atmosphere. As a source of nitrogen plasma, a DC plasmatron with vortex stabilization and an expanding channel of the output electrode was used, generating at the output a weakly diverging plasma jet of nitrogen with a diameter of D = 8–10 mm. Depending on the plasma treatment mode, two types of samples were obtained: type M-plasma with a mass average temperature of 4–6 kK; type A-plasma with a mass-average temperature of 7–10 kK. Processing time was 1 min. Longer treatment led to the destruction of the samples.

Since for the synthesis of the TiO$_2$ coating it was necessary to carry out short-term treatment in a low-temperature nitrogen plasma in the presence of oxygen from the surrounding atmosphere, there was a probability of destruction of the sample, a durable (0001) sapphire was used.

### 2.3. Characterization Methods

Microscopic studies were carried out on a JCM-6000 Neoscope II (JEOL, Tokyo, Japan) scanning electron microscope (SEM) equipped with an energy-dispersive X-ray microanalyzer (ERM) and a Solver Pro-M(NT-MDT, Zelenograd, Russia) atomic force microscope (AFM). The root mean square roughness was determined using Nova software supplied with the instrument. X-ray diffraction patterns were recorded on an Empyrean diffractometer (PANalytical, Almelo, The Netherlands) in the Bragg–Brentano geometry. The X-ray diffraction patterns obtained were processed using HighScore Plus v3.0 analytical software (PANalytical, Malvern, UK); phase analysis was performed using the ICSD database (PDF-2). We used radiation from a copper anode (CuK$\alpha$2 = 1.54 Å). The analysis of the hydrophobicity of the surface (measurement of the contact angle $\theta$) of the samples was carried out using the sessile drop technique. Optical visualization was carried out using a digital camera (Dino-Lite AD413T, New Taipei, Taiwan). A 5 µL water drop was applied. Measurements were carried out 30 s after application, in order to achieve a stable state of the drop. The axis of the camera lens was located at the level of the water drop–sample surface interface. The contact angle was determined by the method described in [18]. The measurements were carried out at 5 different points on the surface of the samples (5 times at each point) and in this work the average result is given.

## 3. Results

### 3.1. Superhydrophobicity of ZnO

3.1.1. Samples of Type I: Au Plating

The sample of type I (Figure 2a), according to electron microscopy data, was an ensemble of hexagonal ZnO microrods up to 1 µm in diameter, from the ends of which pointed ZnO nanorods protruded. The length and diameter of the nanorods were 3–4 µm and 100–150 nm, respectively. Thus, a ZnO structure with bimodal roughness is formed. At the initial stage, hexagonal ZnO microrods are formed according to the vapor–liquid–crystal mechanism [19]. Their growth is catalyzed by liquid zinc islands on the substrate. Oxygen molecules diffuse through zinc vapor to the substrate and react with zinc on the substrate to form zinc oxide. Under conditions of low oxygen concentration, by controlling the temperature in the zinc evaporation zone, the diameter of the growing ZnO structures is controlled. A decrease in the diameter of the ZnO rods is achieved by lowering the temperature in the zone of the zinc evaporator at the final stage of the synthesis. Growth of a ZnO nanoneedle from a zinc droplet of a reduced size at the end of a parent microrod is initiated. Further, by gradually decreasing the concentration of zinc in the gas phase, the growth of ZnO nanoneedles with the formation of spikes at the ends is completely blocked. The micromorphology of the surface of the ZnO sample (type I) after the deposition of gold with a thickness of about 60 nm remained practically unchanged (Figure 2b). The presence of gold was detected only in the ERM spectra (Figure 2c).

In the next stage, the hydrophobic properties of the surface of the obtained samples were investigated. For the test sample of Au film on sapphire, the wetting contact angle $\theta$ was 63° (Figure 3a), which agrees with the data [11]. For a type I ZnO sample without Au deposition, the contact angle was 145° (Figure 3b), and it can be argued that metastable state is realized on the surface. The aggregate of ZnO micro and nanorods forms a surface with a complex morphology and allows the achievement of a contact angle of 145° (realizing the "lotus effect" [20,21]). The contact of a water droplet with hydrophilic ZnO (in the usual state) is minimized and the area of contact between the droplet and the air gap filling micropores in the ZnO mass forms the main interface. The surface of type I ZnO can be characterized as hydrophobic. Taking into account that an ideally smooth surface of ZnO is highly hydrophilic (the contact angle of wetting being not more than 5° [1]), and also considering that the angle of wetting of the gaseous phase with the liquid is 180°, the portion of the area of ZnO (type I) directly in contact with a water drop was determined to be 0.18.

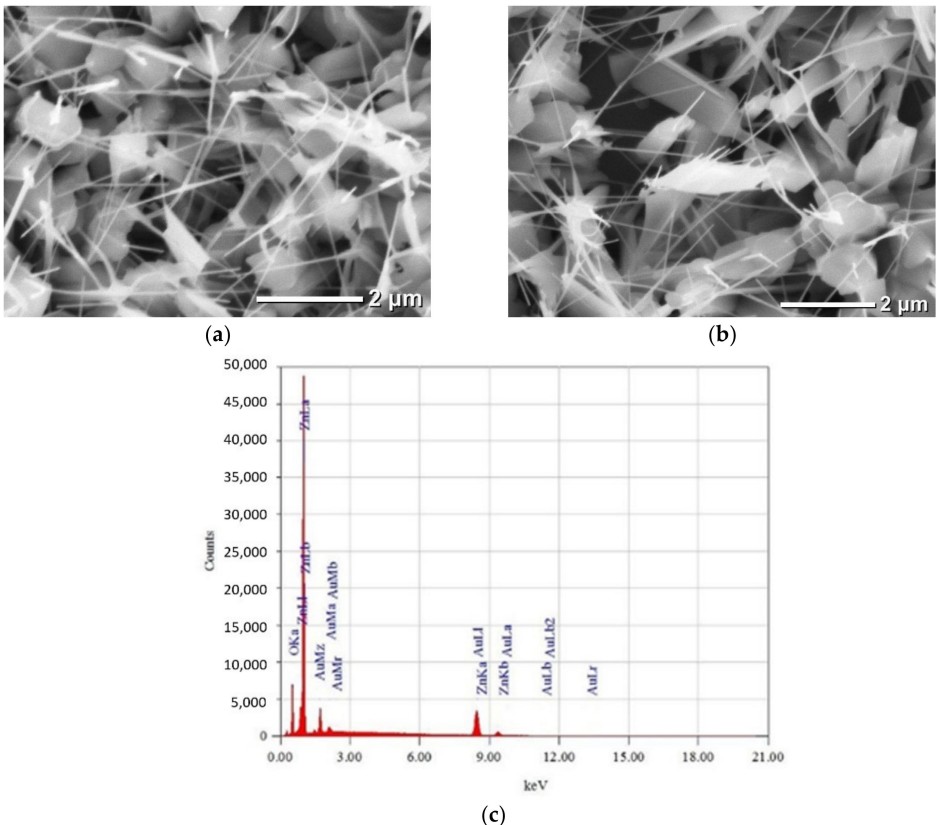

**Figure 2.** Electron microscopic images of the surface and ERM data of type I ZnO samples—Pure (**a**) and with Au (**b,c**).

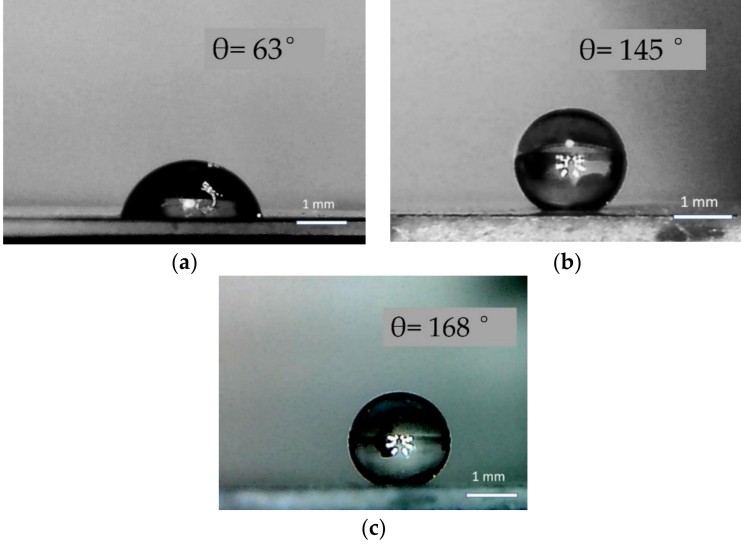

**Figure 3.** Shape of water droplets and contact angles on the surface of a pure Au film (**a**), ZnO samples of type I pure (**b**), and with Au (**c**).

Superhydrophobicity (an increase in the contact angle to 168°) is observed in the area of coverage of a ZnO structure with a gold layer (Figure 3c). It should be noted that similarly high values of the contact angle were observed for ZnO structures of the "coral" type with a polymer coating [9].

In view of the absence of noticeable changes in the ZnO microstructure after gold deposition (Figure 3b,c), it is hardly justified to associate an increase in the wetting contact angle with an increase in the roughness of the side surfaces of ZnO nano- and microrods. It

can be argued that the main factor in enhancing the superhydrophobicity of the surface of a type I sample after Au deposition is the specific interaction of water molecules with gold atoms. An important parameter characterizing the superhydrophobic state is the small sliding angle of the drop. In order for the sample surface to be considered absolutely superhydrophobic, it is necessary to estimate the sliding angles of the water droplet, which should be less than 10°. This is manifested in the study of the rolling of falling water droplets from the surface of the samples. Studies showed that the rolling of a falling drop was observed at tilt angles of more than 30°, which is unacceptable for further applications.

To explain the observed effects, it should be borne in mind that a distinctive feature of the wurtzite structure of ZnO is the presence of polar and non-polar directions. The tip of the nanorods is located along the polar axis C and ends with $Zn^+$ ions on which polar water molecules are easily deposited. The side faces of the ZnO hexagonal rods are formed by non-polar M-planes [22]. Calculations [23] show that water molecules on the M-planes of ZnO aggregate and dissociate, providing a stable hydrophilic state of the surface. In this regard, the expansion of the droplet upon tilting a sample of pure ZnO (type I) can be explained by the tendency toward a decrease in the energy of the water–ZnO interface.

### 3.1.2. Samples of Type II: $Fe_2O_3$ Coating

Samples of type II were synthesized similarly to samples of type I, but Au was used as a catalyst. A periodic Au microstructure on silicon was preliminarily formed. Initially, each element of the Au microstructure is the source of the growth of a plurality of ZnO microrods (Figure 4a) by the vapor–liquid–crystal mechanism [19]. At the next stage, after a sharp decrease in the temperature of the Zn source, ZnO nanoneedles grow from the ends of the microrods (Figure 4a, inset).

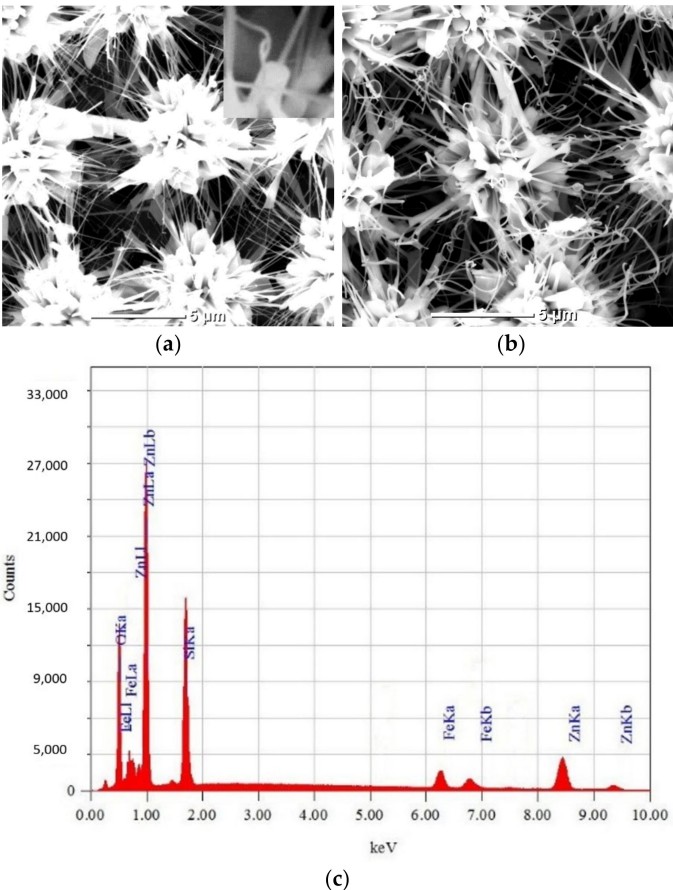

**Figure 4.** Electron microscopic images of the surfaces of samples of ZnO type II pure (**a**) and with $Fe_2O_3$ (**b**). ERM data for a type II ZnO sample with $Fe_2O_3$ (**c**). Inset (**a**): image of a separate ZnO microrod with a nanoneedle.

After the deposition of iron and annealing in air, ZnO nanoneedles are rounded (Figure 4b). At the same time, ZnO microrods retain their shape. The curvature of ZnO nanoneedles is caused by mechanical stresses due to the mismatch of parameters at the interface between zinc oxide and iron (Figure 4c). The thermodynamically stable phase upon annealing in an oxidizing atmosphere at temperatures above 600 °C is $Fe_2O_3$ [24]. However, it is known [25] that at temperatures below 660 °C, two $Fe_2O_3$ phases can exist: hematite and maghemite. Accordingly, annealing of iron under the conditions of our experiment (in atmospheric conditions at a temperature of 700 °C) leads to the formation of a hematite $Fe_2O_3$ coating on the ZnO samples of type II. An annealing time of 1 h is necessary for oxidation of a layer 100-nm thick [26]. The difference between the lattice parameters of ZnO and (hematite) $Fe_2O_3$ is significant: a = 3.25 Å, c = 5.21 Å [27] and a = 5.03 Å, c = 13.74 Å [28], respectively.

The clean surface of a Type II ZnO sample exhibits hydrophobic properties (Figure 5a). Due to the peculiarities of the synthesis, a ZnO surface with multimodal roughness is formed (Figure 4a), which allows the realizing of the "lotus effect" [21]. The contact angle reached a record 173° (Figure 5b) after coating a ZnO type II sample with a layer of $Fe_2O_3$. It should be noted that similarly high values of the contact angle were observed for ZnO structures of the "coral" type (a dendritic structure where each branch breaks down into smaller ones) with a polymer coating [9]. In order for the sample surface to be considered absolutely superhydrophobic, it is necessary to estimate the sliding angles of the water droplet, which should be less than 10°. The studies carried out (Figure 6) demonstrate high water-repellent properties of the surface of a type II sample after the application of a $Fe_2O_3$ layer. Tilting the sample at an angle of 9° leads to the complete roll-off of water droplets falling upon it. A drop of water bounces off a surface of ZnO coated with $Fe_2O_3$ and holds to a clean surface.

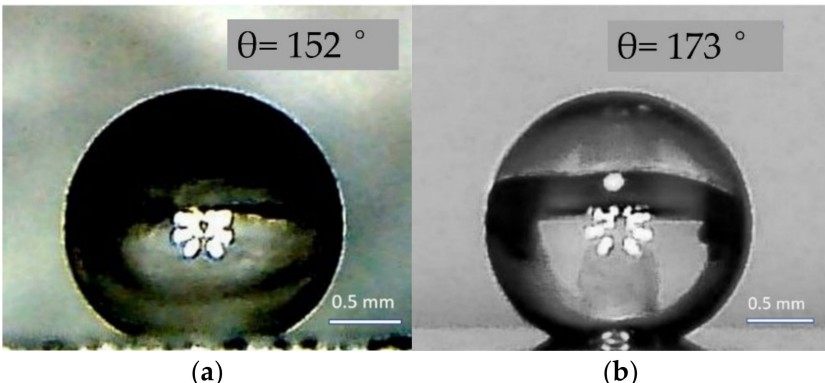

<center>(<b>a</b>)            (<b>b</b>)</center>

**Figure 5.** The shape of water droplets and the magnitude of contact angles on the surfaces of pure type II ZnO (**a**) and with a coating of $Fe_2O_3$ (**b**).

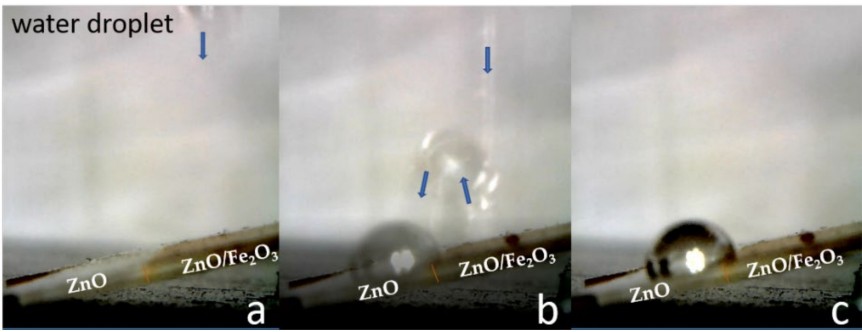

**Figure 6.** Investigation of the sliding angle of a water drop from the surface of a type II ZnO sample covered with a $Fe_2O_3$ layer. The arrows indicate the trajectory of the water drop. Sample inclination angle—9°. Measurement time: (**a**) 0 s, (**b**) 1s, (**c**) 2s.

### 3.2. Hydrophobicity of TiO₂

The correlation between temperature and composition of nitrogen plasma can be carried out in accordance with the kinetic model proposed in [10]. It can be estimated that during the transition from mode M to mode A with an increase in the average mass temperature from 4÷6 to 8÷10 kK, the molecular component of nitrogen plasma significantly decreases. As for ionized molecular ions, their content remains practically unchanged. The atomic component of the nitrogen plasma behaves differently, increasing by several orders of magnitude when passing from mode M to A. The concentration of ionized plasma atoms increases significantly with an increase in the average plasma temperature and reaches a maximum in mode A, as in Figure 7a. It can be seen that the film samples obtained in the M mode are an X-ray amorphous oxidized precipitate. On the contrary, in mode A, a precipitate of the polycrystalline rutile monophase is formed. X-ray diffractometry results are confirmed by scanning electron microscopy data (Figure 7b). The sample in mode A contains faceted crystallites with sizes from 500 nm to 3 μm. The surface of the sample in the M mode contains rounded microstructures without signs of faceting. According to the ERM data, the samples in both processing modes had coatings of oxidized titanium with different nitrogen content. The ratio of N/Ti atoms in sample M is 0.22, and in sample A increases to 0.52. Consequently, the nitrogen content in the type M sample is more than two-times less than in the type A sample.

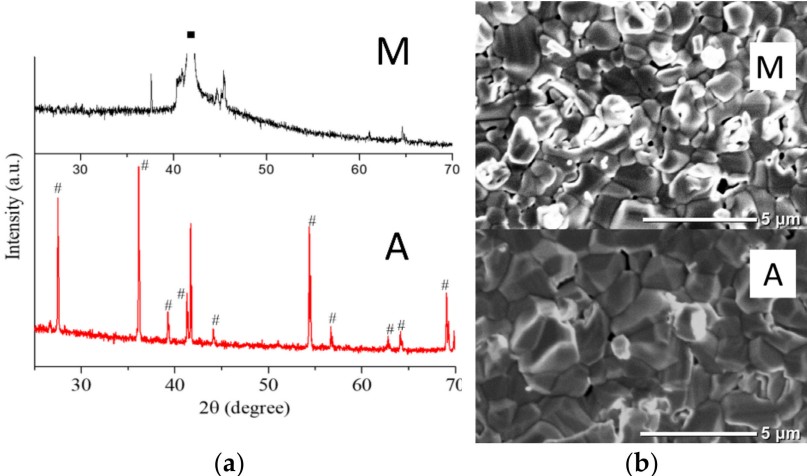

(a)  (b)

**Figure 7.** X-ray diffraction patterns (**a**) and SEM images of the surface (**b**) of samples of titanium oxides of types M and A. Designations: ■—reflections of a sapphire substrate; #—reflections of titanium dioxide (rutile). Card number 98-002-4277.

The high mass-average temperature (4–10 kK) of the applied plasma contributes to the rapid heating of the surrounding atmosphere and, as a consequence, the high activity of oxygen, which in the hot region of the plasma has a highly active atomic structure. Considering that intense titanium oxidation begins at temperatures significantly lower than its nitriding, the main thermochemical processes during plasma treatment in an open atmosphere are titanium oxidation reactions. This explains the predominant oxide phase in the resulting coating. It should also be considered that, in our case, the oxidation process is short-lived and proceeds under acutely non-equilibrium conditions. The influence of the composition of nitrogen plasma is most pronounced in mode A. This is evidenced by both the elemental composition of the coating and X-ray diffraction patterns (Figure 7a). An important factor is the high concentration of atoms and ionized nitrogen atoms, which provide high chemical and diffusion activity. The diffusion and chemical activity of the molecular components of nitrogen, due to their large mass, is much lower. X-ray diffraction patterns and electron microscopy data (Figure 7) indicate the formation of a crystalline rutile deposit only during processing in mode A. Therefore, only in this mode are the conditions necessary for solid-phase crystallization achieved. In addition to the aforementioned

significant atomic component of nitrogen plasma, high diffusion activity is also achieved due to a higher temperature. As for doping with nitrogen as a result of plasma treatment, no increase in the cell parameters of rutile due to the introduction of nitrogen is observed. Instead, this indicates the embedded position of nitrogen atoms in the rutile coating. It can be imagined that active ionized nitrogen atoms form compounds with titanium at the initial stage; however, due to the presence of oxygen atoms and a high temperature, the compounds rapidly decompose with further formation of the oxidized titanium phase. Nitrogen is then displaced to the granule boundaries. Rounded porous formations on the surface of the M-type sample (Figure 7 b) indicate the partial presence of a liquid phase. A lag in the process of oxidation from the titanium melting process and the decrease in the density of titanium in the liquid phase can explain the formation of pores and cavities in the film.

A sample of titanium film on type M sapphire exhibited hydrophilic (contact angle of 73°) properties (Figure 8a), while treatment in mode A (Figure 8b) resulted in a high hydrophobicity (contact angle of 150°) of the sample surface. The results obtained confirm the hydrophobicity of the titanium oxide coating with a monophase structure [5] synthesized in mode A with a significant (the ratio of the atomic components of type A and M plasma being $10^7$) atomic component of the plasma. This feature can be associated with the possible formation of an ordered molecular monolayer of carboxylic acids with pronounced hydrophobic properties on the titanium dioxide surface [6]. For the same reasons, the amorphous surface of titanium oxide synthesized in mode M is characterized by hydrophilicity. As for roughness, its value for the samples processed in the M mode is much higher than in the A mode, due to the high pore density. According to the model [29], an increase in roughness of a hydrophilic surface leads to an increase in the wetting contact angle. Nevertheless, we observe weak hydrophilicity for the titanium oxide sample (type M), which is usually characteristic of low-temperature titanium oxide phases [30]. The important role of nitrogen impurity should be noted. Preliminary calculations show that doping $TiO_2$ clusters with nitrogen increases positive Gibbs adsorption energy and the hydrophobic component, respectively. In this regard, we can definitely speak of a significant increase in the hydrophobicity of titanium oxide samples upon doping with nitrogen.

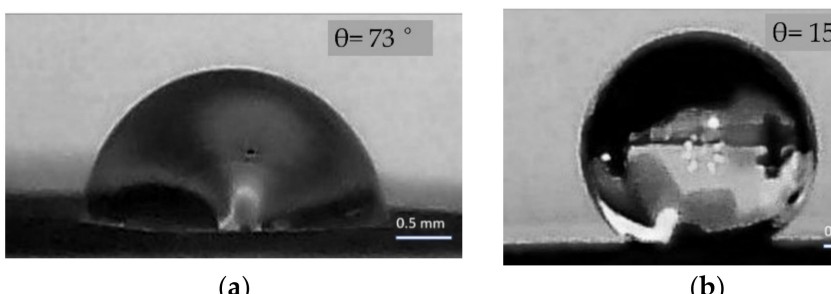

|         |         |
|:-------:|:-------:|
| (**a**) | (**b**) |

**Figure 8.** Forms of water droplets and magnitude of contact angles on the surface of titanium oxides of types M (**a**) and A (**b**).

## 4. Conclusions

In the work presented, new approaches to increasing the hydrophobicity of coatings based on ZnO and $TiO_2$ are considered. Zinc oxide's abundance in nature, low cost, and low toxicity make it an important component of modern electronics. However, there is a problem in increasing the hydrophobicity of the surface of ZnO-based devices. When used to increase hydrophobicity, polymer coatings with good adhesion are distinguished by the technological complexity of application and high economic cost, as well as relatively low decomposition temperatures. On the other hand, in modern technologies, individual materials are not the most in demand, but rather complexes (heterostructures, composites) of materials characterized by increased functionality. ZnO/Au, ZnO/$Fe_2O_3$ systems are efficient photocatalysts. Furthermore, Au is commonly used as electrodes in microelectron-

ics and biosensors. Thus, an increase in hydrophobicity can lead to both an increase in the efficiency of photocatalysis and the durability of devices.

In this work, it is proposed to cover ZnO samples with layers of inorganic materials with a non-polar structure, high melting point, and adhesion to ZnO. Au and hematite $Fe_2O_3$ were used as such materials. Gold contacts are traditionally used in microelectronics, and increasing the hydrophobicity in the Au-ZnO system can increase the life of ZnO-based devices. $Fe_2O_3$/ZnO structures are used as selective gas sensors and the increase in hydrophobicity increases the active area free of wet film adsorbing gases. It was shown that the coating of ZnO structures with bimodal roughness with an Au layer 60-nm thick leads to an increase in the contact angle from 145° to 168°. However, 5-μL water droplets do not roll off the sample surface even at large tilt angles. At the same time, on a clean ZnO surface, the drop slides and the contact area increases; on a gold-coated ZnO surface, sliding and increase in the contact area do not occur (only a significant wetting hysteresis is observed). The maximum superhydrophobicity of a ZnO surface (contact angle of 173°) is achieved after coating with a layer of hematite $Fe_2O_3$. At the same time, the measured sliding angle is 9°, which indicates the high moisture resistance of such a coating. For clean surfaces of Au and hematite $Fe_2O_3$ films, the contact angle wets no more than 70°.

In the case of the use of $TiO_2$ coatings, it is important to have a technology for the controlled formation of a coating with specified hydrophilic or hydrophobic properties. For example, the hydrophilicity of $TiO_2$ coatings is important in implantology, where the main parameters are bio and hemocompatibility. On the contrary, in photocatalysis, it is important to have $TiO_2$ coatings with high hydrophobicity. For titanium oxide coatings, our innovation lies in the proposed method of controlled synthesis using plasma coating technologies with a given crystal structure and a level of doping with nitrogen. It has been shown that the use of nitrogen plasma in an open atmosphere with different compositions (molecular, atomic) makes it possible to obtain both a hydrophilic (contact angle of 73°) and a highly hydrophobic surface (contact angle of 150°). In the first case, under conditions of a high content of molecular nitrogen in the plasma, an X-ray amorphous structure of titanium oxide is formed. In the second, with a high content of atomic nitrogen in the plasma, the rutile monophase is formed, which is the reason for the high hydrophobicity. Moreover, the content of nitrogen impurities in the second case is more than two-times higher, which also contributes to high hydrophobicity.

In synopsis, the main technologies used in this work are: catalytic synthesis of microstructures, magnetron sputtering, heat treatment in atmospheric conditions, and plasma technology. The limitations arising from the application of these technologies are scalability and uniformity of coating, which are exclusively technological problems. Considering the rapid development of magnetron and plasma technologies, one may hope for a quick and positive outcome.

**Author Contributions:** A.E.M., M.K.G., and V.M.K. have carried experiments, participated in writing, reviewing and editing documents, provided financial and technical support, and supervised the entire research process. All authors have read and agreed to the published version of the manuscript.

**Funding:** This research was performed in the frame of state assignments of Ministry of Science and Higher Education of the Russian Federation for FSRC "Crystallography and Photonics" RAS, Federal State Budget Institution JIHT RAS and partially funded by RFBR (the research projects No. 18-29-12099 mk, 20-08-00598).

**Institutional Review Board Statement:** Not applicable.

**Informed Consent Statement:** Not applicable.

**Data Availability Statement:** Data sharing is not applicable to this article.

**Acknowledgments:** The authors are grateful to L.A. Zadorozhnaya for their help in preparing samples. The authors are grateful for technical support from the collective use centers of FSRC "Crystallography and Photonics" RAS.

**Conflicts of Interest:** The authors declare no conflict of interest. The funders had no role in the design of the study; in the collection, analyses, or interpretation of data; in the writing of the manuscript, or in the decision to publish the results.

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
