# Peer review of "New Approaches to Increasing the Superhydrophobicity of Coatings Based on ZnO and TiO2"

_coatings, doi:10.3390/coatings11111369_

Round 1

Reviewer 1 Report

Dear authors,

The manuscript explores the potential of increasing the superhydrophobicity of coatings based on ZnO and TiO2 has been reviewed. As one of the selected reviewers, I read your manuscript carefully. While your research has addressed an important subject, I observed that the manuscript has many severe flaws that must be addressed in a newer version.; some of them have been presented below.

The title is a bit confusing. It does not discuss Fe2O3; however, it appeared 26 times in the text. Consider revising the title appropriately.

The aim of the study at the end of the introduction requires clarification.

I recommend that the authors add a schematic diagram to understand the different coatings used in this study.

Figure 5 should be retaken or presented in the proper way to see the sliding angle properly. No information is given about the water sliding angle of other coatings in this study. These data should be added to characterize the surfaces better. The images used in Figure 5 should be retaken, as the gloss is identified in the middle image, and thus it does not seem to be concurrent and clear. While taking other images, gloss should be avoided.

I could not understand the rationale for adding Figures 9 and 10. It seems that the authors unnecessarily deviate from the original aim of the research. Consider deleting that section; otherwise, clearly define the aim of this research.

Also, consider presenting the extent of time for which the drops in (i) versus (j) might retain on the surface.

Line 61 and 62 Page 2, Typically, polymers have low thermal . . .".

You talked about a "typical behavior" without acknowledging it. Put some references there, please.

Page 8, line 263 "… several micrometers in size". Could you make a more precise quantification of time here?

XPS analysis is missing; the authors are encouraged to perform XPS analysis.

Have you performed a study over time to check the durability and the performance of the modified fabric? This is something that would be more interesting to know and read about.

The reference list should be widened to the latest publications. Here are some examples, even if I'd like Authors to improve the list further.

  1. a) https://doi.org/10.3390/coatings10100943
  2. b) https://doi.org/10.3390/coatings11020151

Please, make your best to compare your materials with results achieved by others. General important question: How your prepared materials are better than many others in this field, including [a] and [b]]? Are they simpler? How much are they simpler? Are they cheaper? How much are they cheaper? Are they less toxic? To what extent? Why do your materials deserve readers' attention?

What are the prospects? Add a separate section or briefly add few sentences in the conclusion section? Also, no research limitation is explained aligned with the research problem. The research limitations describe what dimensions of the problem are excluded by you and your study's boundaries.

The issues mentioned above are for reference only, as I do not recommend this manuscript for publication; thus, no further comments have been added.

I look forward to receiving the revision and reviewing the newer version.

Best of luck.

Author Response

We are grateful to the rewiever for are detailed and have done our best to improve the manuscript.(Corrections are marked in yellow)

Reviewer 2 Report

The study investigates the coating of Zinc oxide with non-polar inorganic coatings to enhance the longevity of the coated surfaces. Although the study is of interest to readers from different fields, there are several issues which need to be addressed from a reader’s point of view. My main concern is the value of 173 degrees reported as well as the sliding angle of 9 degrees: I would be interested to know the magnitudes of contact angles and their respective sliding angles commonly reported on such surfaces. This will greatly help situate the current work in terms of novelty. Also, it is vital that the authors make it as clear as possible why such a high value has been obtained: is it due to the treatment carried out? If so, is there air being trapped beneath the drop such that a Cassie state is achieved (to what do the authors attribute this change to?).

Further comments are elaborated below:

Major comments

I think the abstract needs a better structuring with respect to the results obtained from the study. The highest contact angle achieved (173 deg), described as a ‘record’ on L236 should be the highlight of the study and contrasted with a range of previously reported values on similar surfaces. The authors need to make it clear by how much was the ‘super hydrophobicity’ enhanced, why and what are the implications of such outcomes.

It is not clear from the title and introduction what is/are exactly the coating(s) and what are the substrates being coated. The different constituents involved in the coating process need to be clearly defined and their roles ought to be explicit e.g. what is the purpose of Au, Fe2O3..etc? In addition, has the commonly assumed threshold of 150 degrees been adopted in this study? Are there other criteria for super hydrophobicity in terms e.g. contact angle hysteresis? These need to be specified.

Section 2 needs to be supplemented by figures and schematics to help readers understand what and how were the resulting substrates that were eventually tested.

Although terms such as Lotus leaf is fairly common amongst readers to which this type of research is geared towards, I believe such terms and those like ‘coral’ type need to be briefly referenced in the introduction.

Are the authors associating the hysteresis observed with the 5 ul drop to the bi-modal roughness? Was a parametric study with respect to volume of drop carried out to establish this threshold?

Descriptions are at time qualitative, making use of words like ‘quite natural’, ‘quite easy’, ‘quite enough’…This is not suited for journal articles.

Can the authors mention the targeted applications where the described coating will be more beneficial as opposed to polymeric coatings? Are the methods implemented in the study easy to adopt, less energy intensive as compared to using polymers?

Can the authors justify the selection of drop volume size? Were the measurements of contact angle done in a controlled environment (Humidity and Temperature)- these would very likely influence the measurements given that the drop was on the substrate for a minimum of 30s. Furthermore, all contact angle data reported have no standard errors associated with them, have these tests been repeated? If yes, how many times? Because substrates have been reported to have bimodal roughness, it is very likely that contact angles at different locations diverge to a certain extent.

On L162, how can the argument of Cassie- state be argued in this case. More descriptions are needed? Can the respective area fractions of drop with air and drop with substrate be provided? It could very well be a meta-stable phase between Wenzel and the Cassie-Baxter states. Can the authors comment on this?

Minor but still important comments that should be addressed 

I believe the authors should spell out all chemical compounds at least once before using the respective chemical abbreviations, just like they did with Titanium Oxide (TiO2) in the abstract.

L11: Can the authors be more specific with regards to the type of characteristics which is being contrasted with polymer coatings?

L31 It is better to be more quantitative when reporting this bibliographic analysis, e.g. reporting the rise of literature in the past few decades with regards to ZnO and TiO2.

L42 A better description of the different wetting properties of TiO2 is necessary. It should be clearer as to what influences this property, is it the structure and how it differs from anatase, brookite and rutile?

L54: Gas border? The correct term is three-phase contact point.

L67: How is the active area enhanced? Is it according to the Wenzel model?

L77-84: It is not clear whether this is the process which has been eventually performed in this study. E.g. Nitrogen plasma treatment ‘can’ be used – Has it been used here?

L90-91: This is not a sentence.

L95: ‘no worse’? Please report the resolution of the device accordingly.

L139: What is meant by ‘quite natural’?

L164: Please report contact angles data instead of mentioning the “Lotus effect”.

L178: Scales are needed on these images.

L182: Have these ‘noticeable changes’ been quantified and how?

L229: Scales are needed on these images.

L245: It is not clear what figure 5 depicts. Can the drop be dyed?

L303: Scales are needed on these images.

L307: The contact angles and their ranges need to be reported instead of ‘rather hydrophilic’.

L310: Can the authors elaborate on the atomic component being referred to?

L323: Do the authors attribute high contact angles with the nitrogen treatment?

L338: Has this thickness been quantified or inferred from some analytical techniques?

Author Response

We are grateful to the rewiever for a detailed analysis and have done our best to improve the manuscript 

Reviewer 3 Report

For titanium oxide coatings, an innovation lies in the proposed method of controlled synthesis using plasma coating technologies with a given crystal structure and a level of doping with nitrogen. In the article, it has been shown that the use of nitrogen plasma in an open atmosphere with different compositions (molecular, atomic) makes it possible to obtain both a hydrophilic (contact angle of 73°) and a highly hydrophobic surface. The article is very interesting and at a good level. The research was conducted fairly. I have only a slight sugestions:

  • please provide the percentage composition of the gas mixture of argon and oxygen (line 89)
  • I am asking for the specification or the name of the used research devices manufacturer (eg. JEOL scanning electron microscope - SEM)
  • please describe (in the conclusion)  1-2 sentences on the application of the new solution in practice

Author Response

We are grateful to the rewiewer for a analysis and have done our best to improve the manuscript.

Reviewer 4 Report

The manuscript “New approaches to increasing the superhydrophobicity of coatings based on ZnO and TiO2” reports the development of new methods for performant hydrophobic coatings stabilizing metal oxides such as ZnO and TiO2. Indeed, high moisture resistance and resistance to biofouling / contaminants are challenging aspects for the development of devices based on these oxides. For ZnO, the new approach was to cover ZnO samples with layers of inorganic materials with a non-polar structure, high melting point and adhesion to ZnO (Au, hematite Fe2O3). This strategy was employed  with some success: increase of the contact angle was obtained, but no rolling of the drop upon tilting on the side. For TiO2 samples, the innovative employed method was to perform the controlled synthesis of the substrates: using plasma coating technologies with a given crystal structure and a level of doping with nitrogen. This yielded surfaces with properties ranging from hydrophilic to highly hydrophobic.

This experimental work seems to have been well made and this report may be interesting to the community specializing in this field. However, the manuscript is not yet ready for publication. Improvement in structure of the paper is needed. Hereafter are listed suggestions for improvement.

- The structure of the introduction needs to be improved: main topic, previous works, what is this work is about.

- Please state in the introduction, what is the point in reporting together your results on ZnO and TiO2 in the same publication, as they are different oxides with different behaviors. It is not clear.

- Line 36-40: please insert the information on the crystalline phase for ZnO, that is missing.

- Line 44-50: At this point, the most common crystalline phase encountered in the TiO2-based devices, should be mentioned and focused on.

- Section 2: please add titles “Preparation of X samples…” “characterization methods” for a better understanding.

- Section 2: Please explain why you chose to investigate the photocatalytic decomposition of oleic acid film.

- Figure 8 belongs to Supplementary Information. What is this source? Please provide more details in section 2.

- Figure 9 and related explanations are not clear enough. Please improve.

- Introduction, abstract and conclusion contains similar information. Please minimize repeated information.

- Perspectives of the work: can you please comment on the feasibility of using (expensive) Au for coatings of devices that need to be, as much as possible, cost-efficient?

Author Response

We are grateful to the reviewer for a detailed analysis and have done our best to improve the manuscript

Round 2

Reviewer 1 Report

The authors have improved the manuscript as per the reviewer's guidance; therefore, the manuscript might be considered for publication.

Author Response

We are grateful to the rewiever for the analysis.

Reviewer 2 Report

I would first like to thank the authors in attempting to address my initial comments. However, I feel that the manuscript can and needs to be improved. I hope the comments below serve as a guide for improvements:

  1. The title needs to be as succinct as possible and the parentheses needs to be avoided.
  2. L19-20 of the abstract: Do the authors solely attribute this as being the result of such high contact angle? Are quantifiable measurements described in text?
  3. L27-28: Roughness increases and contact angle increases. This seem a lot like the Wenzel state. Is this the case?

In response to the answers given by the authors, I would have appreciated if the answers were more explicit and less vague. It is not convenient from my point of view to look out for the modifications made in text e.g. “we have given a more specific description”: where and on which lines?

  • What is the criterion with regards to sliding angle?
  • Which external factors are being referred to when contact angle measurements are being made after 30 seconds?
  • I am still not clear about the state of wetting. Is it the metastable, Cassie or Wenzel? How was the 18% obtained- any assumptions made during the calculations?
  • Microscopic investigations were used to quantify roughness? Which technique was adopted? Can the authors report values e.g. Root mean square so that readers can judge whether there were any changes if any?
  • I was suggesting using dyed liquid to improve the contrast (blue usually does the trick).
  • The Schematic needs some labelling with arrows. There are no captions? What does the ‘N’ in TiO2 (N) refer to?

Author Response

We are grateful to the rewiever for a detailed analysis.

Round 3

Reviewer 2 Report

After these rounds of revisions, the manuscript has been improved significantly and is now relatively clearer and more concise.